# The Nexus between Economic Complexity and Energy Consumption under the Context of Sustainable Environment: Evidence from the LMC Countries

**DOI:** 10.3390/ijerph18010124

**Published:** 2020-12-27

**Authors:** Hongbo Liu, Shuanglu Liang, Qingbo Cui

**Affiliations:** School of Economics, Yunnan University, Kunming 650106, China; hongboliu@ynu.edu.cn (H.L.); cqb_ynu@ynu.edu.cn (Q.C.)

**Keywords:** economic complexity indicator (ECI), energy consumption, sustainable environment, Lancang-Mekong Cooperation (LMC), Panel Vector Autoregression (PVAR)

## Abstract

The wide application of various energy resources in economic development is allegedly responsible for deepening environmental deterioration in terms of increasing pollution emissions and other negative consequences including climate change. This current work investigates the interdependent correlation between energy consumption (both fossil fuel energy consumption and renewable energy consumption) and economic complexity among Lancang-Mekong Cooperation (hereafter LMC) countries, from 1991 to 2017. As for empirical analysis, a panel vector autoregression (PVAR) model was employed. Outcomes of this research confirm the existence of a unidirectional relationship between energy consumption and economic complexity index. It is verified that renewable energy usage is a possible alternative to traditional energy and is able to increase economic complexity. This current research proposed to contribute as a pioneering exploration on LMC countries by adding original observations into existing studies. Finally, we will discuss policy implications of this work.

## 1. Introduction

According to a report released by the United Nation on October, 2018, that the world could be on the brink of a climate change disaster if immediate actions are not made. Based on a recent prediction released by U.S. Energy Information Administration (EIA), with the prevailing energy consumption rate, the world is expecting a 50% increase in energy usage by 2050, led by growth in Asia. Energy is indispensable to the functioning of human activities worldwide, nonetheless, increasing energy consumption, especially the non-renewable energy consumption, which has led to severe environmental concerns. Compared to renewable energy resources, fossil fuels in the form of crude oil, coal, along with natural gas, are more commonly adopted for energy resources in developing countries. The production process of fossil fuels is more harmful to the environment which is deemed to increase CO_2_ emissions and deteriorate the environment. One of the major methods to deal with such environmental damage is adopting renewable energy, for instance, solar energy, wind energy and hydropower etc., instead of relying too much on non-renewable energy such as natural gas or coal [1,2]. Unlike non-renewable energy, most renewable energy produces little or nearly zero greenhouse gas emissions.

The transformation from the traditional economy to the green economy is on the top of policymakers’ agendas, and is proposed to evoke a transition of economic operations worldwide [3]. Past decades have witnessed the steady growth of the world’s energy demand and consumption [4,5,6]. Developing countries and emerging markets are developing in an accelerating rate with rapid population growth and industrialization. For developing countries to progress in a track of sustainable path, it is imperative for them to adopt cleaner alternatives for energy consumption in order to reduce climate change effects as well as pollution emission [7].

As a traditional energy source, non-renewable energy served for the development of human beings with a relatively long history. On the contrary, in comparative to the non-renewable energy usage, renewable energy technologies are relatively new, which indicates that they are unable to serve the society at a cost-effective level, especially among developing countries [7,8,9]. All six members of LMC are developing countries, due to their specific geographical location, they possess abundant of natural resources. But as a result of technology or capital restriction, they could not exploit them and make a significant renewable energy contribution to the power system as developed countries do.

The Lancang-Mekong River is one of the world’s major river, ranked second after Amazon in respect to biodiversity. China, Laos, Myanmar, Vietnam, Thailand and Cambodia are six countries alongside this river, with a population amounts to 72 million live within the Mekong basin (as is demonstrated in Figure 1). LMC as a transnational organization encompasses the aforementioned six countries alongside Lancang Mekong river. LMC proposes to contribute in the area of social and political issues, sustainable development among its member countries, along with culture communication [10]. Among the aforementioned objectives, sustainable development is of key significance. Countries of this region have witnessed enormous economic growth, while the side-effects of this increase such as inequality and environmental pollution have also evoked attention. As is demonstrated in Figure 1, countries alongside the Lancang-Mekong river have the opportunity to use renewable energy owing to its congenital rivers dropping variance. The demand for renewable energy has encouraged the construction of hydropower facilities in the Lancang-Mekong river valley [11]. During this process, trans-boundary cooperation among its members will yield a win-win outcome to all collaborators in terms of both economic development and environmental protection.

First introduced by Hausmann and Hidalgo in year 2009, Economic Complexity specified the idea of the multiplicity of advantageous intellect deposit in certain country [12,13]. Economic Complexity is therefore explained as the composition of the productive yield and the arrangements that emerge to absorb and associate proficiency in a country. Through calculating the mix of products that a certain country is capable of producing, it is possible to get access to its Economic Complexity Index [12,13]. First provoked as an enrichment of export diversity, ECI is designed to measure the capability of an economy indirectly by looking at the mix of products that nation exports. However, based on the researches of Eric Kemp-Benedict and Penny Mealy, ECI is orthogonal to export diversity [14,15], which confirms that ECI is not an impeccable design. On the other hand, it is argued that ECI captures significant information concerning the process of economic development in essence, and is capable of supporting the discussion we proposed to make.

The correlation between Economic Complexity and energy consumption can be summarized into four theorems: Neutrality hypothesis, Growth hypothesis, Conservation hypothesis, as well as Feedback hypothesis [1,8,16,17,18]. Among which, no causal relationship between economic complexity and energy consumption indicates neutrality hypothesis which means economic complexity and energy consumption are independent with each other [19]; Growth hypothesis leads to a uni-directional correlation running from energy consumption to economic complexity, implying that innovative sources of energy should be adopted; the existing of a uni-directional causality from economic complexity to energy consumption refers to conservative hypothesis; Feedback hypothesis implies the existence of a bi-directional causal relationship between economic complexity and energy consumption [20,21,22,23,24,25]. 

Existing studies in the empirical literature have been inconclusive in exploring the causality between economic complexity and energy consumption (neither renewable energy consumption nor non-renewable energy consumption). This current research proposed to fill a gap in existing literature with a concentration on the LMC member countries by using Panel Vector-Autoregressive models. What is more, it is expected to contribute to the existing literature in terms of scrutinizing the relationship of Economic Complexity and energy consumption among LMC member countries.

The rest of the paper is structured as follows: Section 2 refers to the literature review. Section 3 presents the methodology framework, data description can be observed in Section 4, Section 5 presents empirical analysis, while Section 6 demonstrates discussion and Section 7 presents relevant conclusion and policy implications.

## 2. Literature Review

The relationship between energy consumption and economic development as well as energy consumption and economic complexity has attracted comprehensive attention from scholars to implement empirical analysis. Analysis of the former literature reveals ambiguous and controversial empirical outcomes. Literature review of this work is grouped into two research strands which were examining the aforementioned topics of environmental economics: first, the indicators we are adopting, including energy consumption, economic complexity; second, the methodologies used in relevant research.

### 2.1. Literature on Economic Development and Energy Consumption

Researches concerning energy consumption and economic development have been explored by numerous studies in former literature. In some studies, the concept of energy consumption was investigated separately from renewable energy consumption aspect, along with non-renewable energy consumption perspective [5,9,26,27]. In precise, Kahia and his colleagues observed a long-term equilibrium correlation between economic growth and renewable energy use and non-renewable energy use among eleven MENA countries from 1080 to 2012 [28]. Furthermore, the development of energy consumption had strong causal relationship to the economic growth as well as the development condition in the latest literature [3], in which, energy consumption was divided into renewable energy consumption and non-renewable consumption. In addition, a study focused on MIST countries including Mexico, Indonesia, South Korea as well as Turkey, revealed the existence of a long-term causal relationship between renewable and non-renewable energy use [29]. This research suggested that nuclear energy a viable approach to enhance energy security and promote sustainable energy economy.

In respect of researches empirically explored the renewable energy consumption exclusively, Adrienne Ohler and Ian Fetters examined the causal relationship between electricity generated from renewable energy sources and economic growth across 20 OECD countries from 1990 to 2008, the results from a panel error correction framework demonstrated the existing of a bidirectional causal relationship between real GDP and aggregate renewable generated from electricity [30]. Chinese scholar Yiping Fang, studied the how economic welfare influences renewable energy consumption in China from 1978 to 2008, by using a multivariate OLS methodology. This research disclosed that an increase in renewable energy usage will contribute to economic welfare development in China [31]. Unlike the above-mentioned research which discovered a uni-directional relationship between concerned variables, Usama Al-mulali identified a bi-directional relationship in the long run between GDP growth and renewable energy usage [32]. Besides, a more up-to-date research originated by Emrah Kocak observed a bi-directional causal relation and confirmed the “Feedback Hypothesis” among Black Sea and Balkan countries [33]. Kocak examined the renewable energy consumption and economic development nexus in 9 concerning countries, by using a heterogeneous panel causality approach from 1990 to 2012. What is more, a study concentrated on six newly industrialized countries revealed a cointegrated relationship between real GDP and renewable energy consumption [34]. In this research, Destek examined the causal relationship between concerned variables by employing an asymmetric causality mechanism, time ranging from 1971 to 2011. 

Different from the aforementioned researches, there exists studies applying Economic Complexity Index as a proxy of economic development condition. As an effective variable to explain fluctuations in country development and economic growth, Economic Complexity Index (ECI) had obtained significant focus among researchers and policy makers from all over the globe [12,14,35,36,37,38,39,40]. According to former literature, nations with a relatively higher ECI index demonstrates similar export baskets with those other countries with a high ECI, which tend to be countries with advanced development status and be able to export products that are relatively more technologically complicated [3]. 

### 2.2. Literature on Panel Vector Autoregressive Model

Md.Samsul Alam indicated in his research that through the methodology of VECM (Vector Error Correction Model) and robust panel cointegration tests framework, it is able to certify that a significant long-run equilibrium relationship among economic growth, oil consumption, finance, trade openness and CO_2_ emissions can be observed in 18 developing countries [41]. Similar methodology was utilized by Hasan Ertygrul in analyzing the influence of trade openness on global CO_2_ emissions for the top 10 emitters among developing countries [42]. As for developed country groups, Tsangyao Chang verified the existence of a bi-directional causal relationship between economic growth and renewable energy from 1990 to 2011 across G7 countries [7]. A heterogeneous panel cointegration test was employed to testify the relationship between renewable energy consumption, economic development (indexed by real GDP and real gross fixed capital formation), as well as the labor force in six Central American nations, time ranges from 1980 to 2006 [16].

Besides the analysis among country groups, there are some methods that are employed on single country analysis. A research concerning the United States revealed that CO_2_ emissions levels are negatively related to renewable energy usage by adopting cointegration and Granger-causality test [43]. Another study concerning the relationship between economic development and electricity usage in China used VAR and VECM model for exploration [44]. A research using time-frequency analysis on France illustrated none robust causal relationship between greenhouse gas emissions and trade openness which confirms the ‘neutral hypothesis’ of the target country [45]. ARDL is another methodology available for single country analysis in the relevant research area, Eyup Dogan found that through this method, it is possible to support the existence of the growth hypothesis in Turkey [20] (see Table 1).

The correlation between economic development and energy consumption have attracted intensive attention in the past three decades, however, analysis of the former literature reveals ambiguous and controversial empirical outcomes. This current research proposed to fill a gap in existing literature with a concentration on the LMC member countries by using Panel Vector-Autoregressive models. 

To conclude, according to the literature summary in Table 1, there is few literatures available concerning economic complexity and energy consumptions regime under the context of sustainable environment. Therefore, this current work is expected to fill the above-mentioned gap. The contributions of this current work to existing literature include: first, we adopt the structural equation methodology technique to investigate the significant relationship between energy consumption and economic complexity under the context of sustainable environment. Second, the deep exploration on such topic in terms of LMC countries was the first time, to the best of our knowledge. Lastly, the work proposed to fill a gap of existing single country researches, for their limitations of reducing the power of unit root and cointegration [26].

## 3. Methodology 

To observe the nexus between Energy Consumption and Economic Complexity in terms of sustainable environment development, a Panel Data Vector Auto-regressive (hereafter PVAR) methodology was adopted. To the best of our knowledge, this specific methodology had not been applied to the subject of energy consumption and economic complexity among LMC countries by so far. PVAR is a scientific research approach entitled with such advantages as: it facilitates the combination of existing VAR method with the panel data approach, to be more specific, it considers all concerning variables in the equation as endogenous factors, which facilitates unobserved individual heterogeneity [49,50]. Furthermore, through PVAR approach, we are able to conquer the issues generated by using granger causality analysis or Vector Error Correction model individually [51,52]. PVAR model enables all variables being considered to be treated as interdependent and endogenous, besides, it is able to model how shocks are transmitted among different countries [53,54].

A general PVAR model can be illustrated as the following equation:(1)$$Y_{it}=Y_{it-1}A_{1}+Y_{it-2}A_{2}+\cdots +Y_{it-p+1}A_{p-1}+Y_{it-p}A_{p}+X_{it}B+\mu_{it}+\varepsilon_{it}$$
(2)$$i\in \left\{1,2,\ldots ,N\right\},\;t\in \left\{1,2,\ldots ,\;T_{i}\right\}$$

The above equation is a i-variate PVAR model of order t, with panel-specific fixed effects, where, $Y_{it}$ is a (1 × i) vector of dependent variables; $X_{it}$ is a (1 × l) vector of exogenous covariates; $\mu_{it}$ and $\varepsilon_{it}$ are (1 × i) vectors of dependent variable-specific fixed-effect and idiosyncratic errors, respectively. The (i × i) matrices (A_1_, A_2_ …, A_p−1_, A_p_) and the (l × i) matrix B are parameters to be estimated. It is assumed that the equations have such characteristics as [55]:(3)$$E\left[e_{it}\right]=0,\;E\left[e_{it}^{’}e_{it}\right]=\sum^{\;},\;E\left[e_{it}^{’}e_{it}\right]=0\;\mathrm{for}\;\mathrm{all}\;t>\;s$$

It is possible to estimate the above parameters through fixed effects or, alternately, independently of the fixed effects after some transformation, using ordinary least squares (OLS) equation-by-equation.

To accomplish those research objectives, this current study designed a second order panel VAR model as follows:
(4)$$Z_{it}=\Gamma_{0}+\Gamma_{1}Z_{it-1}+\Gamma_{2}Z_{it-2}+\mu_{i}+d_{c,t}+\varepsilon_{t}$$
where $Z_{it}$ is a four-variable vector (lnEC, lnECI, lnEXP,lnTRADE), using i to index countries and t to index time, Γ is the parameters and ε is white noise the error term. EC means energy consumption (both renewable energy consumption and fossil fuel consumption will be considered), ECI refers to Economic Complexity Index, EXP is export diversification, TRADE means trade margins which will be represented by extensive trade margins and intensive trade margins, respectively.

In order to utilize a VAR model into panel data analysis, it is proposed to impose restrictions, to make sure that the specific econometric designations are in accordance with each cross-sectional units, in this current case, member countries of LMC cooperation [56]. Therefore, diagnostic investigations such as normality, functional form serial correlation as well as heteroscedasticity analysis are performed to guarantee the reliability of this current study [57]. Maddala and Wu test is proposed to examine the unit root of this research, additionally, in order to decide the lag-order selection, both general Coefficient Determination (CD) operation, and Hasen’s J statistic (J) procedure are conducted. The comprehensive Coefficient Determination (CD), Hansen’s J statistics (J), p-value, MBIC, MAIC, as well as MQIC are calculated to determine lag-order.

## 4. Data

To analyze the nexus between energy consumption and Economic Complexity under the background of sustainable environment, six countries alongside Lancang Mekong river were considered, namely China, Laos, Mymmar, Thailand, Cambodia, Vietnam. Time period ranged from 1991 to 2017, variables include Renewable energy consumption (% of total final energy consumption), Fossil energy consumption, Economic complexity. Renewable energy consumption represents the presentation of renewable energy consumption in total final energy consumption. ECI indicates the knowledge intensity embedded in one economy, it can be measured through considering the knowledge intensity of the products it exports, a higher value of ECI represents an economy with more sophisticated and knowledge intensive production. The value of ECI is calculated through trade data from the UN Comtrade Database, the data of economic complexity comes from Penn World Table version 9.1 [3,12,13,40].

According to Table 2, the total set of data table in our research comprised a sample of 932 observations, which indicated the suitability for adopting Panel Vector Auto-regression model. Besides, the above data summary table demonstrates that the standard deviation is smaller than the mean value, which is suitable for further data analysis [37,58,59].

## 5. Empirical Analysis

We adopted the maximum available data for energy consumption and economic complexity covering from 1991 to 2017, and panel vector autoregression (PVAR) models are used to testify whether the interactions between the concerning variables are bidirectional empirically.

### 5.1. Model Specification

$$\left[\begin{array}{c} {EC}_{t}\\ {ECI}_{t}\\ {EXP}_{t}\\ {TRADE}_{t} \end{array}\right]=\left[\begin{array}{c} \alpha_{1}\\ \alpha_{2}\\ \alpha_{3}\\ \alpha_{4} \end{array}\right]+\left[\begin{array}{cccc} A_{11,1} & A_{12,1} & A_{13,1} & A_{14,1}\\ A_{21,1} & A_{22,1} & A_{23,1} & A_{24,1}\\ A_{31,1} & A_{32,1} & A_{33,1} & A_{34,1}\\ A_{41,1} & A_{42,1} & A_{43,1} & A_{44,1} \end{array}\right]*\left[\begin{array}{c} {EC}_{t-1}\\ {ECI}_{t-1}\\ {EXP}_{t-1}\\ {TRADE}_{t-1} \end{array}\right]+\ldots \;+\left[\begin{array}{cccc} A_{11,w} & A_{12,w} & A_{13,w} & A_{14,w}\\ A_{21,w} & A_{22,w} & A_{23,w} & A_{24,w}\\ A_{31,w} & A_{32,w} & A_{33,w} & A_{34,w}\\ A_{41,w} & A_{42,w} & A_{43,w} & A_{44,w} \end{array}\right]*\left[\begin{array}{c} {EC}_{t-w}\\ {ECI}_{t-w}\\ {EXP}_{t-w}\\ {TRADE}_{t-w} \end{array}\right]+\left[\begin{array}{c} \varepsilon_{1t}\\ \varepsilon_{2t}\\ \varepsilon_{3t}\\ \varepsilon_{4t} \end{array}\right]$$$$+\left[\begin{array}{cccc} A_{11,k} & A_{12,k} & A_{13,k} & A_{14,k}\\ A_{21,k} & A_{22,k} & A_{23,k} & A_{24,k}\\ A_{31,k} & A_{32,k} & A_{33,k} & A_{34,k}\\ A_{41,k} & A_{42,k} & A_{43,k} & A_{44,k} \end{array}\right]*\left[\begin{array}{c} {EC}_{t-k}\\ {ECI}_{t-k}\\ {EXP}_{t-k}\\ {TRADE}_{t-k} \end{array}\right]$$
where EC means energy consumption (both renewable energy usage and traditional energy consumption will be studied), ECI refers to Economic Complexity Index, EXP is export diversification, TRADE means trade margins which will be represented by extensive trade margins and intensive trade margins accordingly. $A_{i,j}$ are polynomials in the lag operator, *ε*_i*t*_ are error-correction terms which are assumed to be random and uncorrelated with mean zero. The following $H_{01}$ to $H_{03}$ are the assumptions the research is focusing on:

$H_{01}$:$\;A_{12,1}=A_{12,2}=\ldots =A_{12,k}=0$, meaning energy consumption is unable to Granger cause economic complexity. 

$H_{02}$:$\;A_{21,1}=A_{21,2}=\ldots =A_{21,k}=0$, referring to export diversification does not Granger cause energy consumption. 

$H_{03}$:$\;A_{13,1}=A_{13,2}=\ldots =A_{13,k}=0$, indicating trade margins does not Granger cause Energy consumption.

The similar is true for other variables.

### 5.2. Unit Root Test 

In order to examine the stationarity of the concerning variables, it is proposed to perform unit root tests before we proceed to panel data estimation [60]. Dickey-Fuller (DF), Augmented Dickey-Fuller (ADF), and Phillips-Perron (PP) are the most popular methods adopted for unit root tests among scholars [16,61,62,63,64]. Based on our data condition, it is suitable to use Maddala and Wu test to examine the existing of unit root among variables.

The unit root test we performed above (Table 3) indicated that the variables in logarithms and the first difference with both non-trend and trend are I(1), therefore, these variables are stationary. Furthermore, the comprehensive Coefficient Determination (CD), Hansen’s J statistics (J), p-value, MBIC, MAIC, as well as MQIC were calculated to determine lag-order [62,63,64,65].

### 5.3. Lag Optimum Test

In order to decide the lag-order selection, both general coefficient determination (CD) operation, and Hasen’s J statistic (J) procedure were conducted. A maximum of four lag was used, including 160 observations, with six panels and an average number T of 17.000, the results are as follows in Table 4.

According to the result of lag length selection model, concerning equations do not include more than a single lag of energy consumption and Economic Complexity Index since the first-order lag demonstrated the smallest criteria.

### 5.4. Results

This section provides the outcomes of panel vector autoregression model, Eigenvalue stability condition, and the analysis of Granger causality Wald test, which can be observed in Table 5. 

The outcomes of PVAR analysis revealed that the increasing economic complexity leads to lower renewable energy consumption while increases fossil fuel energy consumption. On the contrary, renewable energy consumption is expected to boost economic complexity but fossil fuel energy consumption demonstrates the opposite function. Besides, export margin exhibited positive connections between both renewable energy usage and fossil fuel usage. 

Afterward, Eigenvalue stability test (Table 6) was performed in order to scrutinize the stability condition of the PVAR model we examined above.

The Eigenvalue test performed above demonstrated that the PVAR model we built in this research is stable, because all four eigenvalues are inside or on the edge of the unity circle as shown in Table 7.

As the VAR model belongs to the regime of an atheoretical model, which means it is unable to interpret the examined parameters [66,67,68,69,70]. Consequently, it is suggested to concentrate on analyzing impulse response functions and causality investigation. Generally speaking, the Impulse Response Functions (IRF) are capable of evaluating the influence of a certain variable’s shock on the current and future values of endogenous variables while keeping irrelevant shocks mute. However, this technique exerts possible issues of correlation between the residuals in the system. Therefore, in order to remove the possible obstacles of correlation, it is suggested to adopt a shock orthogonalization by using the Cholesky decomposition to isolate the prevailing elements from residuals.

The impulse-response function (demonstrated above in Figure 2) illustrated the causal effects among variables in the short run, medium term, and long term. The Cholesky procedure was used to compute the impulse-response function, the procedure was performed repeatedly for 1000 times to calculate the 5th and 95th percentiles of the impulse responses. 

The issues causing by fossil fuel adoption (including environmental deterioration as well as increasing greenhouse gas emission) are evoking serious social attention in LMC countries. To overcome the above-mentioned debatable issue, it is urgent to introduce renewable energy usage and encourage sustainable economic growth by developing complex economic growth.

## 6. Discussions 

The wide adoption of various energy resources in economic development are deepening environmental deterioration [71]. Extensive comprehension of the relationship between energy consumption and economic development are significant to policy makers in order to make effective environmental policies [21]. The current work investigates the interdependent relationship between energy consumption (both fossil fuel energy consumption and renewable energy consumption) and Economic Complexity among Lancang-Mekong Cooperation (LMC) countries, time ranges from 1991 to 2017. To achieve this purpose, a Panel Vector Auto-regression (PVAR) model was introduced. Outcomes of empirical analysis confirms the presence of a uni-directional relationship between energy consumption and economic complexity index. It is verified that renewable energy usage is a possible alternative to traditional energy and be able to increase economic complexity at the same time contributes to green development. However, in comparative to the non-renewable energy usage, renewable energy technologies are relatively new, which indicates that they are unable to serve the society at a cost-effective level [5,8,20,72], especially among developing countries such as LMC member countries. This current research proposed to contribute as a pioneering exploration on LMC countries by adding original observations to existing studies. 

## 7. Conclusions and Policy Implications

According to the empirical analysis presented in this current work, this paper provides substantial values to policy makers: the existence of an interdependent relationship between economic complexity and renewable energy consumption and nonrenewable encourage the usage of renewable energy more widely, and application of energy conservation policies among LMC countries. Furthermore, readjustment of industrial structure is proposed to increase economic complexity and ensure sustainable economic development. Besides, since hydroelectric power is one of the major energy sources in this region, it is suggested to attract investment to develop technologies that facilitate cleaner energy technologies such as hydroelectric schemes and installation to optimize energy usage efficiency, and exploit more environmentally friendly alternative energy resources. At last, the percentage of other renewable energy constitutions such as wind, nuclear energy, as well as solar should be considered to increase the renewable energy mix in LMC member countries. To conclude, it is suggested for LMC countries to further their commitment as well as cooperate in renewable energy technologies to achieve sustainable environment development.

Through an innovative and data-driven approach, the current research shed new light on controlling environmental degradation and green industrialization among LMC countries and has several distinct implications for the development of sustainable industrial strategy to exert beneficial effect on the environmental quality in these countries. Further researches are encouraged to study the correlation between economic complexity and renewable energy usage by the types of energy, for example, wind energy, solar energy and nuclear energy usage. Besides, due to the limitation of data availability, this current work failed to achieve a more robust result. Therefore, it is rewarding for future researches to do profound analysis by adopting up-to-dated data from various sources in relevant academic realm.

## Figures and Tables

**Figure 1 ijerph-18-00124-f001:**
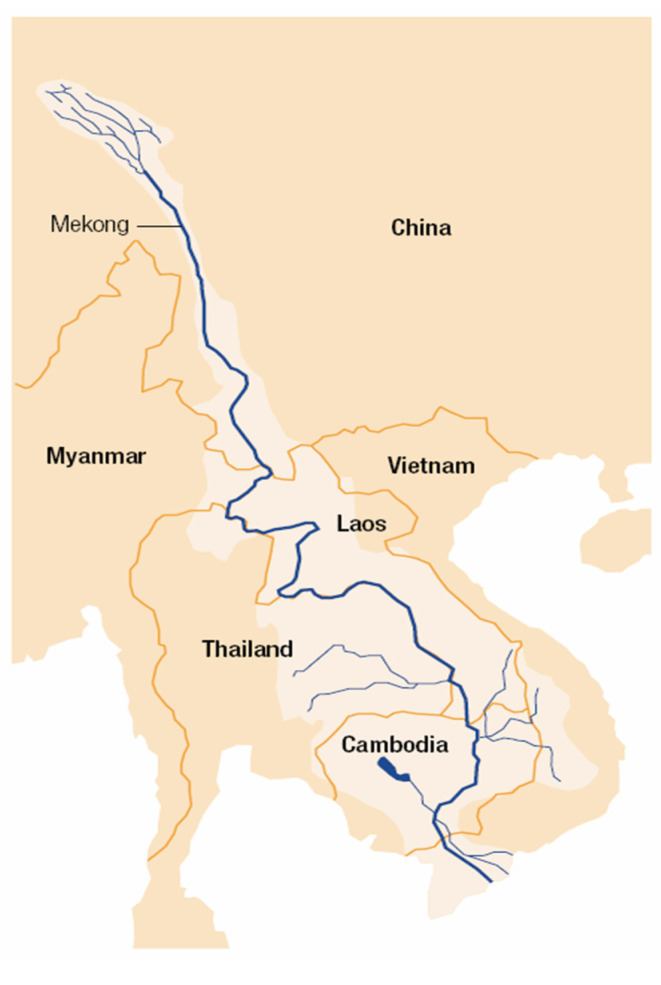
Member countries of LMC.

**Figure 2 ijerph-18-00124-f002:**
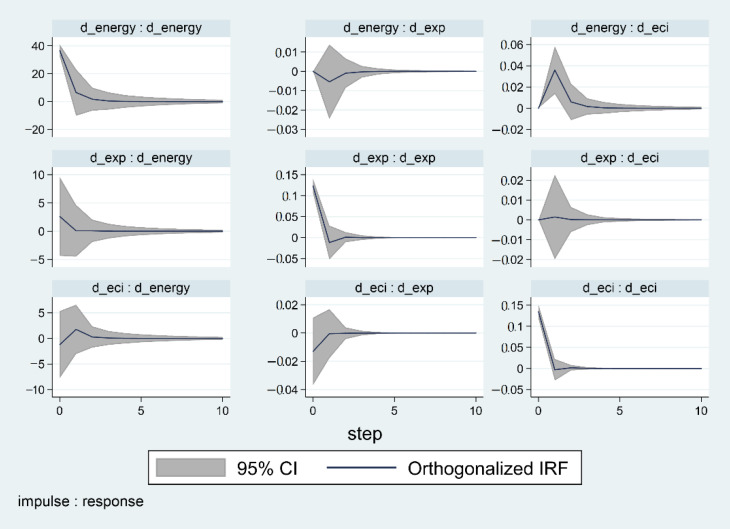
Impulse-response function.

**Table 1 ijerph-18-00124-t001:** Literature review summary.

Authors	Time Period	Country	Methodology	Variables	Empirical Findings	Reference
Dogan Eyup	1990–2012	Turkey	Granger causality test, VECM	Economic growth, renewable energy consumption, nonrenewable energy consumption	Feedback consumption between NRELC and GR	[20]
Syed Ali Raza, Nida Shah	1991–2016	G7 countries	FMOLS, DOLS	CO_2_ emission, GDP, export, import, trade, renewable energy consumption	Support EKC hypothesis	[46]
Nicholas Apergis, James Payne	1980–2004	Central America	Panel cointegration	GDP, energy usage, labor force, capital formation	Support growth hypothesis	[21]
Chi Zhang, Kaile Zhou	1978–2016	China	ARDL, VAR, ECM, OLS	GDP, electricity consumption	Interaction between electricity consumption and economic growth	[44]
Mohammad Jaforullah	1965–2012	United States	Granger causality test	CO_2_ emission, nuclear energy consumption, renewable energy consumption, real GDP, real price of energy	Renewable energy decreases CO_2_ emission	[43]
Medhdi Ben Jebli, Slim Ben Youssef	1980–2010	OECD countries	FMOLS, DOLS	CO_2_ emission, trade, renewable energy consumption	Renewable energy consumption imports	[47]
Mihai Mutascu	1960–2013	France	Wavelet tool	CO_2_ emission, trade openness	Confirm neutral hypothesis	[45]
Pao, Hsiao-Tien	1971–2005	BRIC countries	Panel causality test	CO_2_ emission, GDP, energy consumption	Bidirection causality between energy and emission	[48]

**Table 2 ijerph-18-00124-t002:** Data summary.

Variable	Observations	Mean	Standard Deviation	Minimum	Maximum
Renewable energy consumption	160	54.29	26.54	11.70	91.12
Fossil energy consumption	135	53.72	26.17	13.81	88.90
Economic complexity	160	−0.47	0.75	−1.48	1.16
Export diversification	159	3.20	1.01	1.86	4.85
Extensive margin	159	0.19	0.27	0.002	1.36
Intensive margin	159	3.03	0.97	1.73	4.80

**Table 3 ijerph-18-00124-t003:** Unit root test.

Maddala and Wu-Test				
Variables	Non-TREND		TREND	
	Zt-Bar	*p*-Value	Zt-Bar	*p*-Value
L.EC	1.050	0.902	3.564	0.468
L.ECI	1.230	0.873	4.466	0.347
L.EXP	10.288	0.036	3.481	0.481
L.TRADE	17.171	0.002 **	12.945	0.012
**ΔEC**	28.540	0.001 **	19.432	0.035
**ΔECI**	99.965	0.000 ***	78.068	0.000 ***
**ΔEXP**	32.169	0.000 ***	33.718	0.000 ***
**ΔTRADE**	82.270	0.000 ***	65.915	0.000 ***

Notes: ***, ** denote statistical significance levels of 1% and 5%. The lag length (1) was used. EC = energy consumption. ECI = Economic Complexity Index. EXP = export diversification. TRADE = trade margins.

**Table 4 ijerph-18-00124-t004:** Results of lag length selection procedure.

Lag	CD	J	J *p*-Value	MBIC	MAIC	MQIC
1	0.9991	25.56778	0.0604	−45.515	−6.432	−22.152
2	0.9985	8.077455	0.7842	45.300	−15.989	−27.779
3	0.9988	2.027557	0.9802	−33.514	−13.972	−21.832
4	0.9931	7,897,810	0	7,897,774	7,897,794	7,897,786

**Table 5 ijerph-18-00124-t005:** Results of PVAR analysis.

Response of	Response to			
	Ln_ECI	Ln_EXP	Ln_Renew	Ln_Fossil
Ln_ECI	0.683 ** (4.72)	−1.786 (−0.97)	−0.296 (−0.57)	1.426 (0.37)
Ln_EXP	−0.00664 (−0.41)	0.704 ** (3.22)	0.0836 (1.50)	0.350 (0.84)
Ln_Renew	0.0599 (1.66)	−0.827 (−1.80)	0.916 *** (8.71)	−1.110 (−1.30)
Ln_Fossil	−0.00772 (1.35)	0.159 * (2.04)	0.00391 (0.26)	1.000 *** (7.48)

t statistics in parentheses indicate * *p* < 0.05, ** *p* < 0.01, *** *p* < 0.001. ECI = Economic Complexity Index. EXP = export diversification.

**Table 6 ijerph-18-00124-t006:** Eigenvalue stability test.

Eigenvalue			Graph
**Real**	**Imaginary**	**Modulus**	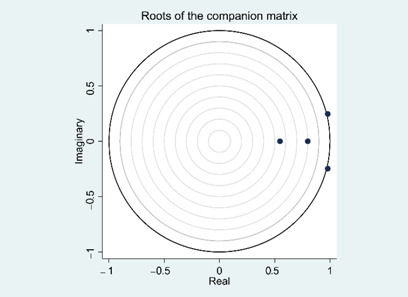
0.979	0.247	1.009994	
0.979	−0.247	1.009994	
0.798	0	0.798	
0.547	0	0.547	

Note: the dots in the above graph indicate eigenvalue.

**Table 7 ijerph-18-00124-t007:** Panel Granger test results.

Equation/Excluded		Chi2	DF	Prob > Chi2
Ln_ECI	Ln_EXP	0.932	1	0.334
	Ln_Renew	0.327	1	0.567
	Ln_Fossil	0.139	1	0.709
	ALL	5.687	3	0.128
Ln_EXP	Ln_ECI	0.166	1	0.683
	Ln_Renew	2.246	1	0.134
	Ln_Fossil	0.702	1	0.402
	ALL	4.898	3	0.179
Ln_Renew	Ln_ECI	2.763	1	0.096
	Ln_EXP	3.250	1	0.071
	Ln_Fossil	1.677	1	0.195
	ALL	8.486	3	0.037
Ln_Fossil	Ln_ECI	1.818	1	0.178
	Ln_EXP	4.173	1	0.041
	Ln_Renew	0.067	1	0.795
	ALL	11.131	3	0.011

ECI = Economic Complexity Index. EXP = export diversification.

## Data Availability

Restrictions apply to the availability of these data. Data was obtained from UN Comtrade Data-base and Penn World. and are available https://growthlab.cid.harvard.edu/files/growthlab/files/atlas_2013_part1.pdf with the permission of UN Comtrade Database and Penn World.

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
