# Peer review of "The Nexus between Economic Complexity and Energy Consumption under the Context of Sustainable Environment: Evidence from the LMC Countries"

_ijerph, 2020, doi:10.3390/ijerph18010124_

Round 1
Reviewer 1 Report
The revised version can be accepted for publication
Author Response
Dear reviewer
Thank you for your work and encouragement
Reviewer 2 Report
Many of the references are dated and need to be updated (eg: global energy projections). Some discussion in Section 6 is unsupported and based on dated information (eg: new text in lines 348 - 350 about renewables). This has major implications for the conclusions. Extensive editing for grammatical errors needed.
Author Response
Dear reviewer,
Thank you for pointing out the mistakes that many people including ourselves would neglect. Discussion concerning global energy projection is rewritten from line 29 to line 31, supported with latest information. We have deleted some dated references and added more updated ones, for example in line 90, line 276, as well as line 290. It really helps us to improve the quality of this manuscript.
Besides, as for the unsupported statement in section 6, we add up-to-date reference to support it from line 349 to line 350.
Lastly, grammatical errors are double checked by both authors and a professional agency. Corrections are made accordingly in the revised manuscript.
Thanks again for your responsible revision.
Reviewer 3 Report
Dear Authors,
I have no more comments to the paper.
Author Response
Dear reviewer,
Thank you for your work and encouragement
This manuscript is a resubmission of an earlier submission. The following is a list of the peer review reports and author responses from that submission.
Round 1
Reviewer 1 Report
The topic has certain significance. However, it requires much overhauling before this work could be considered for publication in this journal. 1. The introduction is poorly motivated. The authors’ emphases on economic complexity and sustainable environment but only few lines are added on these directions. Most part is only dedicated to energy use only. Additionally, the contribution and justification of this study need more explanations in the introduction section. 2. The literature review is also too loose, which is impossible to focus on the topic of the article. The authors are suggested to conduct more accurate literature analysis and add more recent studies. 3. The current research gap based on review should be added at the end of literature review. After reviewing the literature, the study could analyze the shortcomings of the previous research and could pointed out the innovation of this study 4. The rationality of control variables is not provided. Merely stating the names is not sufficient. 5. The “empirical section” just states the results. The discussion is missing. Without providing economic rationale of findings, and comparison with previous studies it is just an econometric display. The authors should improve this part carefully. 6. The paper fails in building the implications of the research. The reason being the results are presented only in terms of relationship based on model estimation, however lacks in discussion of the result. Therefore, it is recommended to rewrite the conclusion section, and expand policy implications based on findings. Additionally, the findings are written in very causal manner. The authors are advised write the current findings in more clear and coherent way. 7. The heading of last section should be “Conclusions and Policy Implications”. 8. The paper needs significant improvements in terms of scholarly language.Author Response
Dear Reviewer,
Thank you for providing us a new perspective to improve this manuscript. We value every suggestion you made, here are our responses:
- The introduction is poorly motivated. The authors’ emphases on economic complexity and sustainable environment but only few lines are added on these directions. Most part is only dedicated to energy use only. Additionally, the contribution and justification of this study need more explanations in the introduction section.
Thank you for your kind remind, for this comment, we did additional descriptions in the revised manuscript from line 81 to line 88, and added references to support it.
- The literature review is also too loose, which is impossible to focus on the topic of the article. The authors are suggested to conduct more accurate literature analysis and add more recent studies.
Revision is done accordingly in the revised manuscript on page 5, page 4, and more recent studies are provided
- The current research gap based on review should be added at the end of literature review. After reviewing the literature, the study could analyze the shortcomings of the previous research and could pointed out the innovation of this study
Research gap is discussed on page 5, at the end of literature review, besides, relevant comments are added
- The rationality of control variables is not provided. Merely stating the names is not sufficient.
The rationality of the major control variables is introduced in “introduction” section,
- The “empirical section” just states the results. The discussion is missing. Without providing economic rationale of findings, and comparison with previous studies it is just an econometric display. The authors should improve this part carefully.
The above mentioned issues is seriously considered, and additional discussion is added in the empirical section from line 305 to 310.
- The paper fails in building the implications of the research. The reason being the results are presented only in terms of relationship based on model estimation, however lacks in discussion of the result. Therefore, it is recommended to rewrite the conclusion section, and expand policy implications based on findings. Additionally, the findings are written in very causal manner. The authors are advised write the current findings in more clear and coherent way.
Thank you for your comments, we have rewritten the conclusion section accordingly, with academic language, besides, policy implications based on empirical findings are added.
- The heading of last section should be “Conclusions and Policy Implications”.
Due correction is made in the revised manuscript on page 11
- The paper needs significant improvements in terms of scholarly language.
As for this issue, we have referred to a professional agency for further procession
To conclude, the reviewer’s suggestions encouraged us to read more literature, and be able to improve this manuscript in various directions, thanks again.
Reviewer 2 Report
The question addressed in this paper is of significance and will be of widespread interest. However, there is significant repetition in the paper and the quality of the writing often impedes the understanding of what is being presented. While the methodology and analysis is adequate, the data used lacks the detail and robustness needed to adequately support the conclusions. The paper needs extensive editing to improve the quality of the writing and additional data or finer detail to support the analysis.
Author Response
The question addressed in this paper is of significance and will be of widespread interest. However, there is significant repetition in the paper and the quality of the writing often impedes the understanding of what is being presented. While the methodology and analysis is adequate, the data used lacks the detail and robustness needed to adequately support the conclusions. The paper needs extensive editing to improve the quality of the writing and additional data or finer detail to support the analysis.
Responses: Thank you so much for your comments, which are very helpful to improve this manuscript.
First, the writing of this paper is improved by a professional agency, and repetition rate is decreased accordingly. Secondly, as for the issue concerning data analysis, due to limited data availability, this is the best we can achieve by so far. Therefore, we have pointed out the disadvantage of this perspective in the revised manuscript from line 366 to line 368.
Reviewer 3 Report
Dear Authors,
in my opinion, this paper is very original and worth publishing.
I would like to formulate three suggestions.
- My key point is clarification of the Economic Complexity and EC Index. There is a strong need to stress that the Economic Complexity Index tries to measure capabilities indirectly by looking at the mix of products that countries export. It is not a perfect design but it works (page 3, at the top). In this context, the limitations of this index should be clearly indicated.
- Discussions and policy implications are the weakest part of your study. There is a strong need to broaden the conclusion "It is verified that renewable energy usage is a possible alternative to traditional energy and be able to increase economic complexity at the same time contributes to sustainable development".
- What are the limitations for renewable energy usage?
- Under what assumptions is there a possible alternative?
- There is a no clear relationship in this sentence between economic complexity index and sustainable development.
Besides, did the specificity of the selected countries influence the research results (LMC member countries) ?
I hope my comments will help to improve the paper.
Author Response
Dear Reviewer,
Your comments are very much helpful and deeply appreciated. You made it clear and workable to improve this manuscript.
- My key point is clarification of the Economic Complexity and EC Index. There is a strong need to stress that the Economic Complexity Index tries to measure capabilities indirectly by looking at the mix of products that countries export. It is not a perfect design but it works (page 3, at the top). In this context, the limitations of this index should be clearly indicated.
Response 1: your recommendation is well accepted and the limitations of Economics Complexity are added in the revised manuscript on page 3 from line 80 to line 86, besides, extra references are provided.
- Discussions and policy implications are the weakest part of your study. There is a strong need to broaden the conclusion "It is verified that renewable energy usage is a possible alternative to traditional energy and be able to increase economic complexity at the same time contributes to sustainable development".
- What are the limitations for renewable energy usage?
- Under what assumptions is there a possible alternative?
- There is a no clear relationship in this sentence between economic complexity index and sustainable development.
Response 2: the limitations of renewable energy usage are supplemented on page 12 in the revised manuscript, besides, the discussion and policy implication section is re-organized accordingly.
Besides, your review encouraged me to do a more responsible job in the future both as an author and a reviewer. Due to time limitation, the correction process does not go well as imagined, but still, thanks for your reviewing.